# The Therapeutic Effect of EZH2 Inhibitors in Targeting Human Papillomavirus Associated Cervical Cancer

**DOI:** 10.3390/cimb47120990

**Published:** 2025-11-27

**Authors:** Dora Vidalina, Lucy Ghali, Nick Kassouf, Shuhan Li, Dong Li, Xuesong Wen

**Affiliations:** 1Department of Natural Sciences, Middlesex University London, The Burroughs, London NW4 4BT, UK; dv214@live.mdx.ac.uk (D.V.); l.ghali@mdx.ac.uk (L.G.); n.kassouf@mdx.ac.uk (N.K.); d.li@mdx.ac.uk (D.L.); 2Respiratory Unit, Homerton Hospital, Homerton Row, London E9 6SR, UK; joanna.li6@nhs.net

**Keywords:** EZH2 inhibitor, HPV16, cervical cancer, epigenetics

## Abstract

High-risk human papillomavirus (HPV) is a crucial risk factor in the development of cervical cancer, where epigenetic modifications and epithelial–mesenchymal transition (EMT) processes have been implicated in cancer progression and metastasis. Enhancer of zeste homolog 2 (EZH2), a histone methyltransferase, is frequently overexpressed in HPV-associated cervical cancers and has been linked to tumour progression. However, there is still no consensus on the mechanisms of their action and their effectiveness on HPV-associated cancers. This study aimed to investigate whether EZH2 inhibitors (EPZ6438 and ZLD1039) can be effective in managing cervical cancer with less toxic effects than the conventional chemotherapeutic drug cisplatin. Proliferation assay and flow cytometry results showed that EZH2 inhibitors effectively induced apoptosis and arrested cells in G0/G1 phase in both HPV+ and HPV- cervical cancer cells. Both inhibitors downregulated the expression of EZH2 and HPV16 E6/E7 at mRNA and protein levels whilst upregulating expressions of p53 and Rb and epithelial markers. In summary, both EZH2 inhibitors showed therapeutic potential in comparison to cisplatin based on cellular and molecular readouts. Additionally, EPZ6438 showed a greater efficacy and higher sensitivity towards HPV+ cells, which was further supported by preliminary in vivo results from the chorioallantoic membrane assay.

## 1. Introduction

### 1.1. High Risk Human Papillomavirus and Cervical Cancer

High-risk HPV, belonging to the *Papillomaviridae* family, is one of the most prominent infectious agents causing cancers of the cervix, vagina, anus, vulva, penis, and head and neck (HNC) by infecting both cutaneous and mucosal epithelium [1]. More than 200 sub types of HPV have been identified, which are divided into high-risk types, such as HPV 16, 18, and 31 which are responsible for tumorigenesis, and low-risk types including HPV 1, 6, and 11 which are responsible for anogenital/cutaneous warts, papillomatosis, and Heck’s disease, etc. [2]. Although HPV is a major causative factor of these cancer types, other exogenous, genetic, and viral cofactors could also contribute to carcinogenesis. Exogenous factors include tobacco smoking, usage of hormonal contraceptives over longer periods, high parity, immunosuppression, co-infections (HIV, Chlamydia trachomatis (CT), and herpes simplex virus type-2 (HSV-2)), and certain dietary deficiencies [3]. Persistent infection with high-risk HPV causes 95% of cervical cancer cases, making it the fourth most common cancer globally in woman [4].

### 1.2. HPV Oncoproteins and Tumour Suppressors in Carcinogenesis

Most high-risk HPV viruses encode eight major proteins, six (E1-2, E4-7) situated in the ‘early’ region that are regulatory proteins in function, and two (L1 and L2) in the ‘late’ region that are structural proteins [1,5,6]. When basal cells of the epithelium are infected by HPV, internalisation of the virus is induced by translation of E1 and E2, which are both responsible for recruiting the DNA components for replication and binding to the replication’s origin. From there, proliferation is induced by protein E5, the differentiated basal epithelium cells’ environment is changed by E6 and E7, and cytoplasmic and nuclear integrity is lost with the help of E4. Oncoprotein E6 binds to tumour suppressor protein p53, promoting its degradation by ubiquitination to inhibit apoptosis. Oncoprotein E7 interacts with retinoblastoma protein (pRb), causing its ubiquitination to activate transcription factors crucial for cell cycle progression. In healthy cells, tumour suppressor pathways p53 and Rb supervise DNA damage response [7]. G1 and G2 arrest, apoptosis induction, and genomic stability are mediated by p53, while regulation of arrest into G1 and entry into S phase and stress responses are via the pRb pathway. Through p53 and pRb mediated inhibitions, among other pathways, HPV E6/E7 manipulates growth suppressor evasion, cell death resistance, and induction of angiogenesis. Namely, proteins E6 and E7 are crucial in discriminating between high- and low-risk HPV as degradation and inactivation of p53 and pRb can only occur in high-risk HPV infected cells [8].

### 1.3. HPV-Driven Transformation via Epithelial–Mesenchymal Transition

While persistent high-risk HPV infection is necessary for malignant transformation, additional cellular and molecular changes are required for tumour progression, invasion, and metastasis [9]. Among these, one key mechanism is the epithelial-to-mesenchymal transition (EMT), a process by which epithelial cells lose their cell integrity and stability characteristics and acquire mesenchymal features that enable increased motility and invasiveness. The transformation involves the transition from tightly connected, apical-basal polarised epithelial cells to spindle-shaped, fibroblast-like mesenchymal cells that exhibit reduced adhesiveness and increased migratory capabilities [10]. On a molecular level, the EMT process is supported by the downregulation of epithelial markers, such as E-cadherin, ZO-1, and cytokeratins, and the upregulation of mesenchymal markers, such as N-cadherin, β-catenin, and Vimentin [11]. Various signalling pathways including TGF-β/Smad, Wnt/β-catenin, Notch, and IL-6/STAT3, and Hedgehog pathways have been shown to play crucial roles in initiating and sustaining EMT through distinct transcription factors such as Snail (SNAI1/2), ZEB1/2, and TWIST1 [12,13].

In addition, HPV infection contributes to EMT through inactivation of tumour suppressors p53 and pRb, while promoting TGF-β signalling and the expression of EMT drivers, such as Wnt, Notch, and SMAD [14]. Furthermore, EMT is a dynamic process that exhibits partial or hybrid EMT state, retaining both epithelial and mesenchymal characteristics [12,15]. This plasticity allows cells to dynamically adapt to environmental pressures, evade immune responses, and resist therapeutic agents [16,17]. Hybrid EMT states are now recognised as highly tumorigenic and more likely to drive metastasis compared to cells in a complete mesenchymal state. Importantly, cells undergoing EMT may revert through mesenchymal-to-epithelial transition (MET) at secondary sites to establish macrometastases, highlighting the reversible and adaptive nature of the process [18]. EMT plasticity is also modulated via EZH2 management of the EMT regulators [19].

### 1.4. EZH2—A Potential Oncogenic Driver

Platinum-based drugs usually elicit a sufficient initial response; however, the majority of patients eventually face cancer relapse and develop drug resistance [20,21].

Recently, a novel therapeutic approach of using demethylating agents, such as polycomb repressive complex proteins (PRC1/2), have shown some encouraging clinical potential in tackling this problem [22,23]. One of the members of family of polycomb group genes (PcGs), which are involved in modulating the chromatin structure via repression of the transcription, is an important epigenetic regulator called Enhancer of zeste homolog 2 (EZH2). The initiation polycomb repressive complex 2 (PRC2) mediates the gene silencing by the catalysation of the mo-, di-, and trimethylation of histone H3 at Lys 27 (H3K27me3). EZH2 functions via three types of mechanism: PRC2-dependent H3K27 methylation, PRC2-dependent non-histone protein methylation, and PRC2-independent gene transactivation. Among them, PRC2-dependent H3K27 methylation plays a key role in gene silencing by repressing transcription via mediation of chromatin compaction.

EZH2 acts as an enzymatic catalytic subunit of PRC2 for H3K27me3 and is crucial in cell cycle, cell proliferation, stem cell development (pluripotency and differentiation), and tissue development, but its aberrant expression/mutation is also essential during malignant transformation [24,25]. EZH2 overexpression is positively correlated with tumour size, lymphatic invasion, and poor patient outcomes [26,27,28]. Elevated EZH2 expression has been reported across several malignancies, including renal, prostate, and cervical cancers, while its expression remains low or undetectable in normal tissues [29]. EZH2 inhibition can suppress tumorigenic behaviours by reducing cell proliferation and invasion, inducing autophagy and apoptosis, and causing G1 phase arrest accompanied by increased expression of tumour suppressors such as p16, p53, and p21 [22,30,31,32,33,34]. High-risk HPV involvement in aberrant EZH2 expression of EZH2 is linked by correlation of p16, a tumour suppressor used as a potential marker for HPV infection with an elevated H3K27me3 epigenome in patients with HPV [35,36,37]. Furthermore, a higher expression level of EZH2 has been linked with dysplastic and pre-cancerous cervical lesions, which are characteristics of HPV infection [38].

Among numerous small-molecule inhibitors of EZH2, this study will evaluate two inhibitors of EZH2 methyltransferase activity: EPZ6438 and ZLD1039. EPZ6438 (tazemetostat or E7438) is a SAM-competitive inhibitor that, through competing with co-factor SAM, directly inhibits enzyme activity of EZH2 [39]. EPZ6438 has demonstrated a favourable safety and tolerability profile with mild to moderate adverse effects that can be resolved without compromising treatment effectiveness [40,41], a high oral bioavailability, and good pharmacokinetic properties [42]. EPZ6438 is currently being tested in numerous clinical trials involving B-cell lymphomas, advanced lymphomas, recurrent ovarian cancer, epithelial sarcoma, etc. [43,44]. However, no clinical trials using EZH2 inhibitor treatment on HPV-associated cervical cancer have been reported. ZLD1039 is another potent and selective SAM- EZH2 inhibitor, which has not been so widely investigated [45,46] but has been shown to have concentration-dependent inhibition of PRC2 enzymatic activity against EZH2 in melanoma cells and breast cancer [47,48].

Overexpression of EZH2 is associated with more clinically aggressive cancers, resulting in poor prognosis; however, due to molecularly and clinically distinctions among cancers, the exact role of EZH2 and possible interaction with HPV remains elusive. This study aimed to investigate the effect of EZH2 inhibitors on cervical cancer cells and evaluate whether their therapeutic effects might be associated with the HPV status in cells. Because EZH2 contributes to HPV oncogene expression through epigenetic repression, we hypothesised that EZH2 inhibitors, EPZ6438 and ZLD1039, could reduce HPV oncogenes and EZH2 expressions levels without altering H3. This could result in tumour suppressors’ restoration, hence promoting cell apoptosis. Moreover, due to the association of EZH2 activity with EMT progression, a possible effect from EZH2 inhibition on the EMT process was also hypothesised.

## 2. Materials and Methods

### 2.1. Cell Culture and Maintenance

HPV16+ CaSki and HPV- C33a (ATCC, Manassas, VA, USA; passages 5–20) human cervical cancer cell lines were cultured in Roswell Park Memorial Institute (RPMI) 1640 (1×) medium (Gibco, Carlsbad, CA, USA) supplemented with 10% Foetal Bovine Serum (FBS) (Gibco, Carlsbad, CA, USA) and 1% penicillin–streptomycin (HyClone, Logan, UT, USA). Normal epithelial cell line CRL1790 (ATCC, Manassas, VA, USA; passages 5–15) was cultured in Minimum Essential Medium (MEM) (Gibco, Carlsbad, CA, USA) supplemented with 10% FBS and 1% penicillin–streptomycin. All cells were cultured in a humidified incubator at 37 °C with 5% CO_2_ and sub-cultured using phosphate-buffered saline (PBS) and trypLE express (Thermo Fisher Scientific, Waltham, MA, USA). Cells were counted using Trypan Blue Solution 0.4% (Thermo Fisher Scientific, Waltham, MA, USA) on Luna cell counting slides (Logos Biosystems, Anyang-si, Gyeonggi-do, South Korea) by automated cell counter (LUNA-II). Preservation of cells was carried out in complete medium with 10% dimethylsulfoxide (DMSO) (Thermo Fisher Scientific, Waltham, MA, USA).

### 2.2. Cytotoxicity Assay

The effect of EZH2 inhibitors EPZ6438 (APExBIO, Houston, TX, USA) and ZLD1039 (Tocris, Bristol, UK) and the standard chemotherapeutic drug cisplatin (Tocris, Bristol, UK) on the proliferation of cervical cancer cells was determined by 3-(4,5-Dimethyl-2-thiazolyl)-2,5-diphenyl-2H-tetrazolium bromide (MTT) assay. Cells were seeded at a density of 5000 cells/well (MTT assay linearity was verified across 1000–100,000 cells/well) in 96-well plates; then, following 24 h settlement, cells were treated with a range of drug concentrations (0, 1.25, 2.5, 5, 10, 20, 40, and 80 µM). After drug exposure for 48 h, 10 µL of 5 mg/mL MTT compound (Sigma-Aldrich, St. Louis, MO, USA) was added; then, cells were incubated for 3 h at 37 °C with 5% CO_2_. To terminate the MTT reaction, 100 µL of MTT solvent (0.4% 2N HCl (Sigma-Aldrich, St. Louis, MO, USA) in isopropanol (Thermo Fisher Scientific, Waltham, MA, USA) was added to each well following a 15 min gentle orbital shaking at room temperature in the dark. The absorbance was read using the plate reader FLUOstar Omega (BMG LABTECH, Aylesbury, UK) within 1 h at OD: 570 nm.

### 2.3. Treatment of Cells

Cells were treated for 48 h with determined IC50 concentrations in the complete medium. Cisplatin was dissolved in 0.9% NaCl_2_ and both EPZ6438 and ZLD1039 were dissolved in DMSO. Final DMSO concentration was 0.1%.

### 2.4. Apoptosis Detection Assay

A density of 50,000 cells per well was seeded in 6-well plates. After overnight cell settlement, they were treated with drugs for 48 h. An apoptosis assay was then carried out using an Annexin V-FITC/PI apoptosis staining kit (Thermo Fisher Scientific, Waltham, MA, USA) as per the manufacturer’s instructions. Cells were analysed using a flow cytometer BD Accuri C6 Plus (BD Biosciences, Franklin Lakes, NJ, USA) within 1 h and analysed using BD Accuri C6 Plus software. A total number of 10,000 singlet events was collected for each sample. Gating included exclusion of debris (FSC/SSC), singlet selection (FSC-A vs. FSC-H), and fluorescence-based identification of different stages of apoptotic populations.

### 2.5. Cell Cycle Assay

Distribution of DNA content across cell cycle phases was determined using FxCycle™ PI/RNase staining solution (Invitrogen, Waltham, MA, USA). Following 48 h treatment, 100,000 cells/per sample were resuspended in 500 µL of FxCycle™ PI/RNase staining solution and incubated for 30 min in the dark at room temperature with agitation every 5 min. Samples were measured using a flow cytometer BD Accuri C6 Plus (BD Biosciences, Franklin Lakes, NJ, USA) and analysed using BD Accuri C6 Plus software. A total number of 10,000 singlet events was collected for each sample. Gating included exclusion of debris (FSC/SSC), singlet selection (FSC-A vs. FSC-H), and fluorescence-based separation of cell cycle phases.

### 2.6. Scratch Wound-Healing Assay

A density of 200,000 cells/well was grown until cells had formed a highly confluent layer. A sterile 200 µL pipette tip was used to make a scratch on the cell layer inside a 6-well plate for all samples prior to adding the drug treatments. Migration rate of cells was recorded at 0, 24, and 48 h following drug treatment. Measurements of the width of the scratch were taken at three random positions and were quantified using the INFINITY Software v6.5.0 (Lumenera Corporation, Ottawa, ON, Canada). Wound closure was initially assessed at a 6 h timepoint to confirm migratory activity before proliferation could influence the assay. All cell lines showed measurable wound closure at 6 h under all treatments. However, more pronounced differences were observed at 24 and 48 h timepoints which have been included in the final analysis.

### 2.7. Immunocytochemical Staining (ICC)

Cells were grown on individual sterile coverslip inside a 6-well plate and treated for 48 h. Cells were fixed with 4% paraformaldehyde (Sigma-Aldrich, St. Louis, MO, USA) for 8 min and then exposed to 0.1% Triton-X100 in PBS (Sigma-Aldrich, St. Louis, MO, USA) for 7 min at room temperature. Subsequently, cells were blocked with blocking serum in PBS (PK-6200; Vectastain Elite ABC kit, Newark, CA, USA) for 8 min before they were incubated for 90 min in specific primary antibodies diluted in PBS (Appendix A). The secondary antibody was used for 30 min followed by 20 min tertiary antibody (PK-6200; Vectastain Elite ABC kit, Newark, CA, USA). Finally, TSA-CY5 reagent (NEL745001KT Akoya Biosciences, Marlborough, MA, USA) was added for 5 min before one drop of antifade mounting media with DAPI (H-1200 Vectashield; Newark, CA, USA) was applied. The coverslips were kept at 4 °C overnight in the dark before observation under a fluorescent microscope (Nikon eclipse Ti2, Nikon Instruments, Amstelveen, The Netherlands). Images were processed using NIS Elements 5.11.01 software.

### 2.8. Western Blot Analysis

Following 48 h drug treatment, cells from each sample were scraped, collected using Eppendorf tubes, then lysed in RIPA buffer (Thermo Fisher Scientific, Waltham, MA, USA) with protease inhibitor (Thermo Fisher Scientific, Waltham, MA, USA). The lysates were sonicated three times at 50% amplitude for 2 s with 1 min rest on ice. Bradford assay was performed using bovine serum albumin (BSA) (Bio-Rad, Hercules, CA, USA) stock solution and Bradford Coomassie reagent (Sigma-Aldrich, St. Louis, MO, USA). The absorbance values were measured using the plate reader FLUOstar Omega (BMG LABTECH, Aylesbury, UK) at 595 nm. Molecular ladder precision plus protein standard dual colour 10–250 KDa (Bio-Rad, Hercules, CA, USA) and equivalent amounts of total protein (50 μg) with 2× Laemmli SDS reducing buffer (Alfa Aesar, Haverhill, MA, USA) were electrophoresed in Mini-protein TBX pre-cast gels (15%) (Bio-Rad, Hercules, CA, USA) with 1× running Tris-glycine SDS buffer (Thermo Fisher Scientific, Waltham, MA, USA) (3.03 g Tris, 14.4 g Glycine and 1 g SDS in dH_2_O) at 100 V. The proteins were then transferred electrophoretically to a Trans-Blot Turbo Transfer pack (Bio-Rad, Hercules, CA, USA) using the transblot turbo transfer system (Bio-Rad, Hercules, CA, USA) at 2.5 A, 25 V for 7 min. The membranes were stained in Ponceau S solution stain (Sigma-Aldrich, ST. Louis, MO, USA), rinsed in 0.1% PBST (Sigma-Aldrich, St. Louis, MO, USA), and blocked in 5% BSA in 0.1% PBST for 1 h on a rocking shaker (Ohaus, Parsippany, NJ, USA). Membranes were then incubated overnight at 4 °C on a see-saw shaker with specific primary antibodies diluted in PBS (Appendix A). Membranes were washed with 0.1% PBST following incubation with either goat anti-mouse IgG HRP Conjugate (1:2000; Bio-Rad, Hercules, CA, USA) or anti-rabbit IgG HRP Conjugate (1:3000; Bio-Rad, Hercules, CA, USA) for 1 h at room temperature. The membranes were then visualised using the Pierce ECL (electrochemiluminescence) Western blot substrate (Thermo Fisher Scientific, Waltham, MA, USA) in Odyssey XF Dual-Mode Imaging System (Li-Cor) with Image studio 6.0 programme (National Institutes of Health, Bethesda, MD, USA). Finally, membranes were probed with anti-human beta actin (Appendix A) to control equal loading.

### 2.9. Chorioallantoic Membrane (CAM) Assay

Fertilised chicken eggs (Medeggs, UK) were cleaned with warmed up (43.3–48.9 °C) disinfectant (Brinsea, UK), labelled, and incubated at 37.8 °C with 60–70% humidity in Ova-Easy Advance incubator (Series II; Brinsea, UK) with automatic 180° rotation for 3 days. On day 4, the egg candler (Brinsea, UK) was placed in the dark against the eggshell to locate and mark the air cell. The marked hole was drilled with a scalpel by placing it at 45° and rotating. Using a 20 G syringe needle at a 90° angle, 5 mL of albumen was removed. A strip of scotch tape was placed on the eggshell and a ~1 cm^2^ window was created with a scalpel in the eggshell. The opening was then sealed with transparent tape. Eggs were returned to the incubator with the window side facing up. On day 6, the NuOss collagen block (Collagen Matrix, Franklin Lakes, NJ, USA) was cut into suitable sizes, washed with 70% ethanol, rinsed in PBS, and soaked in RPMI1640. Density of 2,000,000 cervical cancer cells (C33a and CaSki; passage number 10–15) in 30 μL of complete media RPMI1640 were seeded on the cut blocks and incubated for 30 min before adding 1 mL RPMI1640 in the individual Eppendorf tube. Tubes were vented and kept overnight in a humidified incubator at 37 °C with 5% CO_2_. On day 7, viable eggs’ CAM was gently scratched on the smallest blood vessel to induce mild bleeding, and scaffolds were implanted directly onto the bleed spot. Windows were resealed and eggs returned to the incubator. On day 11, implanted scaffolds were treated by applying 30 μL of cisplatin and EPZ6438 (at their IC50 concentrations). On day 13, eggs were chilled on ice for ≥30 min, and viable eggs were opened. Tumour-bearing CAM tissue was excised, washed in PBS, and fixed in 4% PFA for 48 h, then decalcified in 0.5 M EDTA (Sigma Aldrich, St. Louis, MO, USA) (pH 7.2–7.4) for 3 weeks.

### 2.10. Histological Examination

#### 2.10.1. Tissue Processing

Following decalcification, tissues were dehydrated through hourly increasing strength of ethanol (40%, 50%, 70%, 90%, 2 × 100%) and cleared in xylene (Fisher Scientific, Hampton, NH, USA) (2 × 100%). Tissues were embedded in paraffin blocks by gradually infiltrating molten (60 °C) paraffin wax (VWR Chemicals, PA, USA) mixed with xylene: 50:50, 75:25, and 100:0%. Sections were cut at 5 µm in thickness using a rotary microtome (Leica RM2235, Wetzlar, Germany), floated on a 40 °C water bath, and mounted onto slides overnight for drying.

#### 2.10.2. Immunohistochemistry Staining

Sections were deparaffinized in xylene (2 × 5 min) and rehydrated through gradually reducing the strength of ethanol for 3 min each (2 × 100%, 90%, 70%). Endogenous peroxidase activity was blocked using cold 3% hydrogen peroxide (Sigma Aldrich, St. Louis, MO, USA) in methanol (Fisher Scientific, Hampton, NH, USA) for 10 min. Antigen retrieval was performed in boiled citrate buffer (trisodium citrate dihydrate, citric acid monohydrate, and EDTA disodium salt; Fisher Scientific, Hampton, NH, USA) (pH 6.0) for 20 min at room temperature. After two washes with PBS, sections were blocked with 50% normal horse serum (Sigma Aldrich, St. Louis, MO, USA) in PBS for 10 min and then incubated with primary antibody in PBS overnight at 4 °C or for 3 h at room temperature (Appendix A). Sections were washed with PBS and incubated with biotinylated secondary antibody and avidin-biotin complex (Vectastain Elite ABC kit, Newark, CA, USA) for 1 h each. Slides were stained with DAB substrate (Dako, Santa Clara, CA, USA) for up to 10 min, rinsed in dH_2_O, counterstained with Gill’s haematoxylin for 6 min (Sigma Aldrich, St. Louis, MO, USA), differentiated in 1% acid alcohol (Fisher Scientific, Hampton, NH, USA), and washed under running tap water for 5 min. Tissues were then dehydrated through increasing the strength of ethanol (70%, 90%, 2 × 100%) at 3 min for each step and cleared in 2× xylene (5 min each step) before mounting with a DPX mountant (Sigma Aldrich, St. Louis, MO, USA). Microscopic images were processed under a brightfield microscope (Nikon eclipse Ti2) in NIS Elements 5.11.01 software.

### 2.11. RT-qPCR

RNA extraction from collected treated cell pellets was performed following PureLink RNA Mini kit protocol (227957; Invitrogen, Waltham, MA, USA). A total of 200 ng RNA was converted to cDNA by reverse transcription following SuperScript IV Reverse Transcriptase (Invitrogen, Waltham, MA, USA) protocol. qPCR was implemented using the FastStart Essential DNA Green Master (Roche, Indianapolis, IN, USA) in a LightCycler 96 Instrument (Roche, Indianapolis, IN, USA). The primers are listed in Appendix A, with GAPDH as the internal control.

### 2.12. Statistical Analysis

The data were analysed using Minitab (v22.4.0) and GraphPad Prism (v10.5.0). Depending on normality and equal variance tests, data were analysed using ANOVA followed by Tukey’s multiple comparison test, or by Kruskal–Wallis followed by Dunn’s post hoc test for non-parametric data. For pairwise comparisons between cell lines for the same treatment, the Mann–Whitney U test was used for non-parametric data. For all post hoc comparisons, familywise error was controlled using Holm–Bonferroni correction (or Tukey for parametric ANOVA). Effect sizes (η^2^ for ANOVA/Kruskal–Wallis, Cohen’s d or rank-biserial correlation for Mann–Whitney) and exact *p*-values were reported. Data were presented as mean ± standard deviation (SD), with statistical significance defined as *p* ≤ 0.05. Significant statistical differences were indicated as *: *p* ≤ 0.05, **: *p* ≤ 0.01, ***: *p* ≤ 0.001, ****: *p* ≤ 0.0001.

## 3. Results

### 3.1. EZH2 Inhibitors Reduced Viability of Cervical Cancer Cells in a Dose-Dependent Manner

Cytotoxicity of the positive control drug cisplatin and two EZH2 inhibitors was evaluated by MTT assay on two cervical cancer cell lines (HPV- C33a; HPV+ CaSki) and non-cancerous epithelial cell line following 48 h drug exposure (Figure 1). The findings are presented as a percentage change in cell viability from drug-treated viable cells against controls. The cell proliferation rates decreased with the increasing drug concentrations following in which the 50% inhibitory concentrations (IC50) values (with 95% confidence intervals) were determined for all three drugs from two cell lines, respectively. IC50 value metrics and maximal observed inhibition (Emax) for each cell line are provided in Appendix A. Cisplatin IC50 value for HPV+ cells is higher compared to its respective HPV- cells, whereas EPZ6438-treated and ZLD1039-treated cells showed lower IC50 values in HPV+ cells, indicating that HPV+ cells are more sensitive to the EZH2 inhibitor treatment, especially to EPZ6438 (Figure 1D). Statistical significance was observed between cisplatin treated HPV+ and HPV- cells (*p* = 0.0023) and normal cells (*p* = 0.0072), and between normal cells and cancer cells for EPZ6438 (*p* < 0.0005). Additionally, drug treatments were also checked on the non-cancerous epithelial cell line for cytotoxicity with MTT assay (Figure 1) and Annexin V/FITC PI staining by flow cytometry (Appendix A). Data suggests that EPZ6438 has little to no effect on normal cells, whereas ZLD1039 treatment displayed a slight increase in necrotic cells compared to controls.

### 3.2. EZH2 Inhibitors Induce Apoptosis and G0/G1 Arrest in Cervical Cancer Cells

Cell cycle and apoptosis assays were analysed via flow cytometry following 48 h drug treatment on HPV- C33a and HPV+ CaSki cells. The DNA content distribution across cell cycle phases from treated HPV- and HPV+ cells at the 48 h timepoint is displayed in Figure 2A. The mean percentage of cell number in different phases of the cell cycle is shown in Figure 2B. Across two cell lines, both EZH2 inhibitors’ treatment arrested cells in G0/G1 phase, whereas cisplatin significantly induced cell cycle arrest in S phase for HPV- C33a and in G0/G1 for HPV+ CaSki cells. Moreover, cisplatin treatment on both cell lines had a significantly higher percentage of cells in sub-G1 compared with both EZH2 inhibitors (Figure 2C). Following two-way ANOVA analysis, cisplatin treatment caused a significantly higher number of cells to be shown in sub-G1 phase for HPV- C33a cells compared with HPV+ CaSki cells (*p* = 0.027) after exposure to cisplatin, while it was opposite for ZLD1039 treatment (*p* = 0.0235).

Given the findings from the sub-G1 population, the next step was to distinguish different stages of cell death following treatment using a cell apoptosis assay (Figure 3A and Appendix A). Results showed that increased apoptosis was observed in all drug-treated samples in comparison to the controls, and ZLD1039 showed the most significant effect among them (*p* = 0.001) (Figure 3A). Although all drug treatments showed an increase in apoptotic cell population in comparison to HPV- cells, a significant increase in apoptotic cells was found from the HPV+ cell line following ZLD1039 treatment (*p* < 0.0001).

Apoptosis induction following EZH2 inhibition was further confirmed by assessing the protein and mRNA expression of Caspase 3, an apoptosis biomarker. HPV- and HPV+ cervical cancer cells were stained with DAPI for nuclear counter staining and TSA-CY5 for labelling Caspase 3 expression in cells. Western blot was carried out for relative protein abundance to confirm the result. RT-qPCR for relative Caspase 3 mRNA expression was also investigated in comparison to the untreated sample. In Figure 3B, Caspase 3 protein expression seems to be localised in the cytoplasm but also in nuclei for some cells, and ZLD1039 significantly increased Caspase 3 expression in CaSki cells in comparison to other treatments and HPV- C33a cells. Total protein abundance analysis not only showed the increase following ZLD1039 but also a significant difference following cisplatin treatment in HPV- cells (Figure 3C). Although both analyses showed an increase in Caspase 3 protein expression following all treatments, mRNA expression of Caspase 3 was only increased following treatment with EZH2 inhibitors in both cells (Figure 3D).

### 3.3. EZH2 and H3 Protein and mRNA Expression Decreased Following EZH2 Inhibition

Catalysation of methylation of Histone 3 (H3) is the main function of EZH2; therefore, the protein expression of epigenetic change was evaluated for EZH2 and H3 in HPV- and HPV+ cervical cancer cells. Immunocytochemical expression of EZH2 and H3 protein levels is displayed in Figure 4A. The staining pattern and intensity were similar across two cell lines. The presence of both EZH2 and H3 protein was found mostly in the nuclei. EZH2 protein expression (Figure 4B) was reduced following treatment with EZH2 inhibitors, but cisplatin treatment surprisingly showed a significant increase in EZH2 presence in both cell lines. A significant decrease in EZH2 mRNA levels in HPV+ cells was found following both EPZ6438 (*p* = 0.003) and ZLD1039 (*p* = 0.0027) treatment compared with HPV- cells (Figure 4C), while EZH2 mRNA levels significantly decreased in HPV- cells from cisplatin treatment. H3 expression levels were inconsistent between ICC and Western blot results; however, H3 mRNA expression decreased following all treatments in both cell lines.

### 3.4. EZH2 Inhibitors Reduced Expression of HPV Oncogenes E6 and E7 Followed by an Increase in Tumour Suppressors

HPV16 E6 and HPV16 E7 expressions were only analysed for HPV+ CaSki cells, as confirmation of C33a cells not being cross-contaminated with HPV16/18 has been carried out with PCR (Appendix A). HPV16-related oncogenes E6 and E7 were immunostained and then analysed for relative protein expression intensity (Figure 5A). The expression of both proteins was shown as intense staining of the cytoplasm of HPV+ CaSki cells. The relative quantification of E6 and E7 expression showed that both EZH2 inhibitors reduced the expression of HPV oncoproteins, with a significant E6 reduction observed from EPZ6438-treated cells. Western blot analysis was only successful for HPV16 E6, and it showed a reduction in E6 protein expression after treatment with EZH2 inhibitors (Figure 5B). After examining the protein expression levels, further examination was carried out on the gene level. Reversed transcribed cDNA from drug-treated CaSki cells was analysed for relative mRNA expression and compared with the untreated sample (Figure 5C). Both E6 and E7 mRNA levels were reduced following drug treatments.

The effect of the EZH2 inhibitor on tumour suppressor proteins p53 and pRb was evaluated using immunocytochemistry staining, Western blotting, and RT-qPCR (Figure 6). The ICC staining results displayed strong nuclear staining for both proteins, but some cytoplasm staining for p53 was also observed (Figure 6A). The increase in expressions of p53 and pRb was confirmed by mean fluorescence intensity analysis (Figure 6A) and Western blot results (Figure 6B), with significantly increased p53 expression following both EPZ6438 and ZLD1039 treatment in HPV+ cells; however, a similar finding was not seen from HPV- C33a cells. Similarly, an increase in TP53 and Rb1 mRNA expression was observed in both cell lines following treatment with EPZ6438 and ZLD1039, while cisplatin-treated cells displayed a decrease for both tumour suppressors (Figure 6C).

### 3.5. EMT Involvement in Cervical Cancer and Possible Reversal Following EPZ6438 Treatment

The impact of cell migration behaviour was evaluated using the scratch wound–healing assay (Figure 7A). A significant difference in slower wound closure was observed for cisplatin treatment in HPV+ cells (*p* < 0.0001) and in EPZ6438 compared to controls in HPV+ cells (*p* < 0.0473). Migration rate reduction was observed following 48 h treatment with EZH2 inhibitors (Figure 7B). Additionally, HPV+ cells exhibited a much faster scratch closure rate compared to the HPV- cells.

To further confirm EMT involvement, epithelial markers (E-cadherin and ZO-1) and mesenchymal markers (β-catenin and Vimentin) were evaluated. Transmembrane protein E-cadherin exhibited membranous localization with moderate cytoplasmic staining. E-cadherin expression was only observed in HPV- C33a cells with ICC staining, where it showed increased intensity of staining following all treatments, with a significant increase observed for ZLD1039-treated cells (*p* < 0.0001) (Figure 8A). In HPV+ cells, cisplatin and ZLD1039 significantly upregulated mean fluorescence intensity of E-cadherin, whilst total protein and mRNA expressions were upregulated for both EPZ6438 and ZLD1039 treatment (Figure 8B,C). Surprisingly, E-cadherin’s expression was downregulated following cisplatin treatment. Tight junction marker ZO-1 displayed strong membrane-associated staining with weaker cytoplasmic staining and was upregulated in both cell lines at a protein level following all three drug treatments, and it was significantly upregulated for HPV+ cells following EZH2 inhibitors’ treatment at the mRNA level (*p* < 0.0244) (Figure 8).

Two mesenchymal markers, β-catenin and Vimentin, were evaluated at both the protein and mRNA levels (Figure 9). Both markers displayed strong cytoplasmic expression, but β-catenin also displayed membrane staining in some cells (Figure 9A). Western blot failed to detect the presence of β-catenin in the HPV- C33a cell line. ICC fluorescent staining of HPV- cells showed a reduction in β-catenin from all drugs’ treatment, while an increase was seen from ZLD1039-treated cells at the mRNA level (Figure 9A,C). Similarly, in HPV+ cells, treatment with ZLD1039 upregulated β-catenin and Vimentin expression at both the protein and mRNA levels (Figure 9B,C). For both cell lines, cisplatin displayed the highest decrease in β-catenin and Vimentin expression at both the protein and mRNA levels. However, mRNA expressions were inconsistent for ZLD1039-treated HPV- cells and EPZ6438 treatment compared with their protein expression profiles (Figure 9C).

### 3.6. In Vivo CAM Assay Preliminary Validation of the Result from In Vitro Study Following EPZ6438 Treatment

Current results have demonstrated that EPZ6438 has better effectiveness on HPV+ cervical cancer cells (Caski) than ZLD1039-treated cells, especially with respect to cytotoxicity and downregulating mesenchymal markers whilst upregulating epithelial markers. Hence, EPZ6438 was further verified in vivo using the CAM assay. Preliminary study using the CAM model from seeded cervical cancer cells (Caski and C33a) was analysed by IHC. IHC images are displayed in Figure 10. Consistent with the results obtained in vitro at the both protein and mRNA levels, the IHC results detected a tendency towards reduced expression of EZH2 and β-catenin and increased expression of p53 following treatment with EPZ6438 in both samples. However, ZO-1 expression seems upregulated following EPZ6438 treatment, which was not observed from C33a established tissue sections. HPV16 E6 protein intensity in cell nuclei appeared to be reduced following EPZ6438 and cisplatin treatment compared with the control (DMSO 0.1%) treated sample.

## 4. Discussion

High-risk HPV is the leading cause of cervical cancer, and the prevalences of HPV-associated cancer are currently increasing. Recent studies on epigenetic reprogramming have found significant EZH2 overexpression/mutation in different malignancies [29,49]. In this study, EZH2 inhibitors EPZ6438 and ZLD1039 have been investigated for their cytotoxicity, apoptosis induction, and cell cycle arrest following their exposure on HPV+ and HPV- cervical cancer cells. Their impact on these cells was further evaluated at the protein and mRNA expression levels for various epigenetic markers, oncogenes, EMT markers, and tumour suppressors. Results indicate that EZH2 inhibitors, especially EPZ6438, have therapeutic potential by inhibiting tumour cells’ growth whilst enhancing treatment sensitivity on HPV+ cells, inducing apoptosis and cell cycle arrest at G0/G1 phase, reducing cell migration, downregulating HPV oncogenes and epigenetic markers, and upregulating tumour suppressors and epithelial markers. Downregulation of mesenchymal markers by EPZ6438 also suggests a possible reversal of the EMT process; therefore, epigenetic reprogramming could be taking place following EPZ6438’s exposure, which could be another important therapeutic factor along with downregulating HPV oncogenes. The preliminary in vivo CAM assay following EPZ6438 treatment on seeded cervical cancer cells further confirmed the findings from the above-mentioned in vitro studies.

Our findings are consistent with previous studies with respect to the functional role of EZH2 inhibitors on epigenetic effect and limiting cell differentiation of HPV-associated cancers [49,50,51]. The EZH2 inhibitor EPZ6438 has previously demonstrated strong inhibitory effects with a dramatic decrease in global H3K27me3 levels in head and neck squamous cell carcinoma (HNSCC) cells [52], and it has been FDA approved for the treatment of locally advanced or metastatic follicular lymphoma and epithelioid sarcoma [53,54]. A recent study by Tang, Yang and Sun [55] investigated the mechanism through which EZH2 promotes tumour-associated macrophages in HPV16+ cervical cancer cells. To our knowledge, this is the first study to investigate the effect of EPZ6438 treatment on cervical cancer CaSki cells. Results showed reduced cell proliferation and inflammatory response stimulation [55]. Furthermore, ZLD1039 has been shown to suppress tumour growth and metastasis, as well as affect the cell-cycle-related gene expression on melanoma cells [47], but its therapeutic effect on cervical cancer has not been reported. A previous study has investigated the role of EZH2 in the progression of HNSCC using the CAM model [56]. The findings revealed that tumour models generated from seeded HNSCC cells without EZH2 knockdown exhibited more abnormal cellular features and tissue structure including a disrupted basement membrane, enlarged cell nuclei, little cytoplasm, and spindled cellular morphology. In contrast, EZH2 knockdown resulted in tumours with well-differentiated epithelial cells characterised by keratin formation and a less aggressive and more epithelioid phenotype. This finding agrees with our current result from EMT markers following EZH2 inhibitors’ treatment in vitro. The results (Figure 8 and Figure 9) are also consistent with Liu’s group [56] by demonstrating a potential upregulation of epithelial marker E-cadherin and a downregulation of mesenchymal marker of β-catenin. EMT markers’ profiling from the preliminary in vivo CAM model (Figure 10), especially the model generated by HPV+ CaSKi cells, further confirmed our in vitro findings. This indicates that EZH2 inhibitors, such as EPZ6438, could be considered as a potential therapeutic drug in managing high-risk HPV-associated cervical cancer due to its tendency to decrease HPV E6 expression. Indeed, further investigations are warranted to support the current findings by providing more evidence at the cellular and proteomics levels.

Since the initiation and progression of tumour development are primarily driven by the HPV E6 and E7 oncogenes, targeting these oncogenes is considered as one of the most promising strategies for cervical cancer therapy [57]. Our findings showed that both EZH2 inhibitors along with the conventional chemotherapeutic drug cisplatin decreased HPV16 E6 and E7 expression, but EPZ6438 treatment exhibited the least toxic effect among them. Further investigation could explore whether higher treatment doses or a combination of treatments could further reduce HPV16 oncogene expression with a better chemoprotective effect towards normal cells. More context-dependent studies of HPV oncoproteins focusing on tissue-specific signalling, stage of infection, and variations in host gene and microenvironment are needed [58,59]. Nevertheless, downstream functional targets, such as tumour suppressors p53 and Rb, were reactivated (Figure 6) following EZH2 inhibitors’ treatment, which is also confirmed with apoptosis induction and cell cycle arrest in G0/G1 phase with some differences between HPV- and HPV+ cells (Figure 2 and Figure 3). However, previous studies have shown that tumours with mutant p53 accumulate the protein but fail to undergo proper tumour-suppressive pathway processes leading to apoptosis, especially in response to chemotherapy-like cisplatin [60,61,62]. This discrepancy is noticeable between p53 protein accumulation and mRNA downregulation under cisplatin treatment for both cells. Given that the CaSki cell line expresses the wild-type TP53 gene, p53 upregulation likely reflects a response to the treatment, which is also evident with Rb upregulation, G1/G0 arrest, and apoptosis induction, as well as with HPV E6/E7 downregulation. However, the HPV- C33a cell line contains mutant p53, which means that p53 overexpression likely reflects mutant, non-functional p53 that accumulates due to impaired degradation [63]. Therefore, an in-depth study should assess p53 pathway targets, like p21 or Bax, or other genetic investigations such as the reporter gene assay or Next-Generation Sequencing (NGS) could be the next step forward to determine if p53 is mutated or active. This could further clarify the molecular mechanism of cervical carcinogenesis and whether p53 mutation could be playing a role in the effectiveness of EZH2 inhibitors’ treatment and cancer prognosis [59]. Moreover, results of Caspase 3 (cysteine-aspartic acid protease 3) expression were inconsistent between its protein and mRNA levels despite having shown apoptosis induction. Both EZH2 inhibitors displayed an upregulation, more pronounced at mRNA levels, which could likely reflect active transcriptional upregulation with temporarily delayed translation or protein which is rapidly processed to a cleaved form. Additionally, ZLD1039 was reported to activate Caspase 3 in breast cancer [46]. Caspase 3 is one of the key enzymes in the late stage of apoptosis, and it is cleaved to active form in the final execution phase [64]. These results could suggest that total protein is slightly changed, but a cleaved form which is undetected at protein level could be high, indicating active apoptosis. On the other hand, cisplatin-treated cell samples displayed upregulated Caspase 3 at the protein level and downregulation at the mRNA level. This could also indicate post-transcriptional regulation or caspase-independent apoptosis mechanisms [65]. To further distinguish the status of cleaved form, several approaches could be employed including caspase activity assay and using specific antibodies to avoid cross-reactivity [64,66,67].

An emerging number of studies [38,68,69] suggest that epigenetic reprogramming of EMT via EZH2 inhibitors could promote anticancer effects, as EMT induction poses a significant challenge in the management of HPV-associated cancers, severely impacting efforts to extend patient survival. In our study, following the treatment with EZH2 inhibitors, Vimentin expression levels varied among the investigated cell lines. This is not entirely unexpected. Although Vimentin overexpression is associated with increased invasiveness and metastatic potential [70], the EMT process often remains incomplete. Consequently, the absence of certain mesenchymal markers, such as Vimentin and β-catenin, may reflect a partial phenotypic transition rather than a full shift to mesenchymal characteristics [71]. Similar findings have also been reported from other studies by showing the detection of the combination of both epithelial and mesenchymal markers, i.e., only partial loss of E-cadherin was exhibited, or co-expression of both markers [71,72,73]. This could explain the discrepancies in the expressions of mesenchymal and epithelial markers from both cell lines following EZH2 treatments. Specifically, in HPV- C33a cells, ZO-1 protein increased while mRNA decreased for all three drug treatments, E-cadherin was absent at both the mRNA and protein levels, and mRNA levels of β-catenin and protein levels of Vimentin decreased following cisplatin and EPZ6438 but increased following ZLD1039 treatment. Similarly, in HPV+ CaSki cells, at both the protein and mRNA levels, ZO-1 and E-cadherin increased following EZH2 inhibitors’ exposure, while β-catenin and Vimentin decreased post-cisplatin and EPZ6438 treatment. Taken together, these changed patterns of studied EMT markers may indicate cell-line-specific responses and partial EMT states, where some epithelial markers are retained while mesenchymal markers show variable expression levels, which is consistent with the previously published literature [71,74]. These observations suggest a hybrid or partial EMT state, where epithelial markers are upregulated while mesenchymal markers such as Vimentin show variable transcriptional but limited translational changes. This hybrid phenotype is also reflected by a slight β-catenin downregulation in CaSki cells, which indicates an incomplete EMT or partial MET. This is consistent with functional tumour plasticity that allows simultaneous adhesion and motility [75,76]. According to human protein atlas [77], the C33a cell line exhibits no expression of CDH1 gene, which encodes E-cadherin; however, mass spectrometry proteomics data could further confirm its protein expression. Furthermore, loss of E-cadherin expression in cervical cancer can be partially due to DNA methylation [78].

Indeed, the current findings of this study hold promise of using EZH2 inhibitors in treating HPV-associated cervical cancers; however, several limitations should be acknowledged. Firstly, lack of H3K27me3 data restricts us from fully evaluating the epigenetic mechanisms underlying the observed effects. Furthermore, the observed inconsistencies of H3 expressions between ICC and Western blot may be associated with issues from antibody specificity, epitope masking, or differential detection of histone modifications. Future studies directly quantifying H3K27me3 via Western blot or chromatin immunoprecipitation (ChIP) qPCR will be important to investigate these inconsistencies in order to clarify these underlying mechanisms. Secondly, the in vivo validation experiment using the CAM assay was a preliminary study. Expanding this assay to a larger cohort and complementing the immunohistochemical analysis with RT-qPCR would further confirm the results and strengthen the validity of the findings. Additional in-depth epigenetic profiling and expanded in-vivo experimentation will be essential to confirm and extend these results. Moreover, our wound-healing assay did not include proliferation inhibition. Although early wound displacement was confirmed at 6 h, the absence of mitomycin-C or serum deprivation means that proliferation may contribute to wound closure at later timepoints, such as 48 h. Future studies employing proliferation-controlled migration assays are needed to confirm the specific effects of EZH2 inhibitors on cell motility.

Overall, this study demonstrates the therapeutic effect of EZH2 inhibitors on cervical cancer cells. Furthermore, compared with the HPV- C33a cell line, these drugs show more effectiveness in reducing epigenetic markers, reversing EMT markers, and upregulating tumour suppressors for HPV+ Caski cells in vitro. To confirm whether EZH2 inhibitors, especially EPZ6438, have a superior role in managing cervical cancers, especially HPV-associated cervical cancers, further studies incorporating a larger cell-sample size, perhaps even including patient-derived samples, are necessary. Involvement of various molecular pathways has been revealed to differ between HPV- and HPV+ cancers by recent studies [79,80,81]. Investigating the pattern of differential gene expression and verifying the underlying potential mechanism could provide new insights into development of HPV-associated cancers. Specifically, 164 genes were found differentially expressed only in HPV- in comparison with HPV+ and non-cancerous cervical cancer tissues, while there are 1419 differentially expressed genes in both HPV- and HPV+ so far [82]. Considering that EZH2 inhibitors can potentially reverse aberrant DNA patterns, methylation-specific modifications in both HPV- and HPV+ cervical cancer should be analysed to further determine effectiveness of EZH2 inhibitors [83,84,85,86,87]. Identifying specific mechanism changes between HPV- and HPV+ cancers has the potential to provide novel insights into their carcinogenesis, including identifying potential biomarkers, leading to selective targeted therapies.

Moreover, cisplatin resistance has been linked to EZH2 overexpression in different cancers including cervical, ovarian, and breast [88,89,90]. In particular, EZH2 inhibition was shown to induce reversal of cisplatin resistance and to increase cells’ sensitivity to cisplatin in cervical cancer cells [88]. Therefore, exploring the combination of an EZH2 inhibitor with cisplatin, or administering the EZH2 inhibitor as a secondary treatment following cancer recurrence post-cisplatin therapy, may represent a promising therapeutic strategy [91]. The previous published literature has shown positive therapeutic impact on various malignancies in the use of EZH2 inhibitors when they were combined with other chemotherapeutic drugs; for instance, in prostate [92], lung [93], ovarian [94], and breast cancer [95]. Recent studies have also started designing EZH2-based dual inhibitors due to their synergistic antitumour effects with other cancer agents [39,48]. Our findings suggest that HPV+ cervical cancer cells display greater resistance to cisplatin compared to HPV-cells, whilst an increased sensitivity to EZH2 inhibitor EPZ6438 was indicated in vitro. Further investigation is necessary to validate these preclinical results and assess their potential implications for the treatment of other HPV-associated cancer cells.

## 5. Conclusions

This study highlights the therapeutic potential of EZH2 inhibitors—particularly EPZ6438—in HPV-associated cervical cancer cells. These findings suggest that EZH2 inhibitors have greater antitumour activity than the conventional chemotherapeutic agent cisplatin. EPZ6438 demonstrated a better efficacy in inducing G0/G1 cell cycle arrest, promoting apoptosis, and driving epigenetic reprogramming. The preliminary in vivo validation using the CAM assay provides additional support for its enhanced specificity against HPV-positive cervical cancer. Overall, EZH2 inhibitor EPZ6438 emerges as a promising targeted treatment strategy for HPV-associated cervical malignancies, which warrants further study and possible exploration with other HPV-associated malignancies.

## Figures and Tables

**Figure 1 cimb-47-00990-f001:**
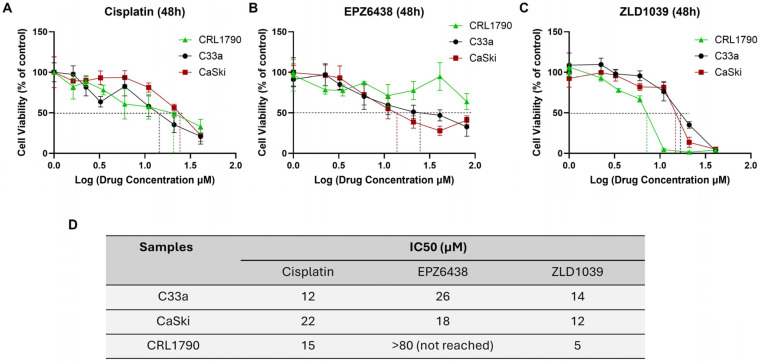
MTT assay results for cytotoxicity of drug treatments. Drug dose–response curves after 48 h for HPV- C33a, HPV+ CaSki, and normal epithelial cell line CRL1790: (**A**) cisplatin, (**B**) EPZ6438, and (**C**) ZLD1039. Data is expressed as log of mean ± SD (n = 3). IC50 values, pinpointed on the graph, were calculated using GraphPad prism. (**D**) Table of mean IC50 value data (n = 3). The IC50 value for EPZ6438 could not be determined within the 0–80 µM range.

**Figure 2 cimb-47-00990-f002:**
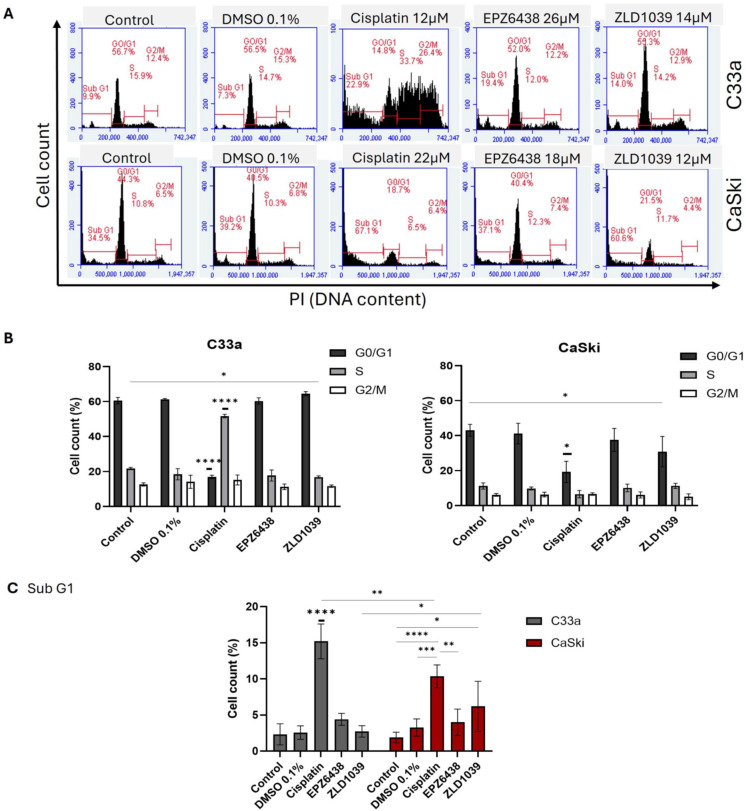
Flow cytometry analysis of cell cycle distribution following 48 h treatment. (**A**) Representative original histogram data to show the distribution of cells in different cell cycle phases. Total events = 10,000. (**B**) The percentage of cell cycle distribution between treatments is shown for G0/G1, S, and G2/M phases. (**C**) The percentage of cells in sub-G1 phase. Data is presented as mean ± SD (n = 3). * *p* ≤ 0.05, ** *p* ≤ 0.01, *** *p* ≤ 0.001, **** *p* ≤ 0.0001. Bold asterisk shown above a single sample bar indicates the significant difference of that sample from the rest of the treatments.

**Figure 3 cimb-47-00990-f003:**
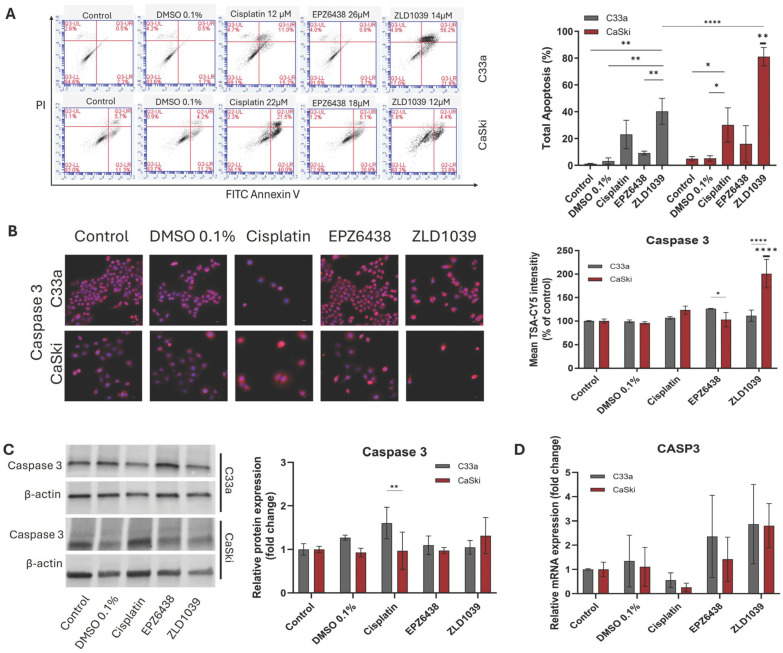
Flow cytometry analysis of cell apoptosis and Caspase 3 expression at both protein and mRNA level after 48 h treatment. (**A**) Representative scatter plots of treated cell lines with Annexin V FITC/PI double staining (left). Cells are distributed into live (LL), early apoptotic (LR), late apoptotic (UR), and necrotic (UL) regions. Total events = 10,000. Analysis of total cell apoptosis (early and late apoptosis) normalised to control (n = 3) (right). (**B**) Immunofluorescence staining of Caspase 3 protein expression (pink) and nuclei (blue) under 400× microscope magnification. Scale = 50 µm. Mean intensity of red fluorescence was normalised to untreated cells (n = 3). (**C**) Representative Western blots’ results of Caspase 3 protein bands. Observed molecular weights are 35 kDa (Caspase 3) and 42 kDa (β-actin). Western blot quantification of Caspase 3 protein expression levels following treatment are shown on the right. Values are expressed as fold change of control (n = 3). (**D**) RT-qPCR analysis of relative CASP3 mRNA expression levels. Values are expressed as fold change of control values. Data is presented as mean ± SD (n = 3). * *p* ≤ 0.05, ** *p* ≤ 0.01, **** *p* ≤ 0.0001. Bold asterisk shown above a single sample bar indicates the significant difference of that sample from the rest of the treatments.

**Figure 4 cimb-47-00990-f004:**
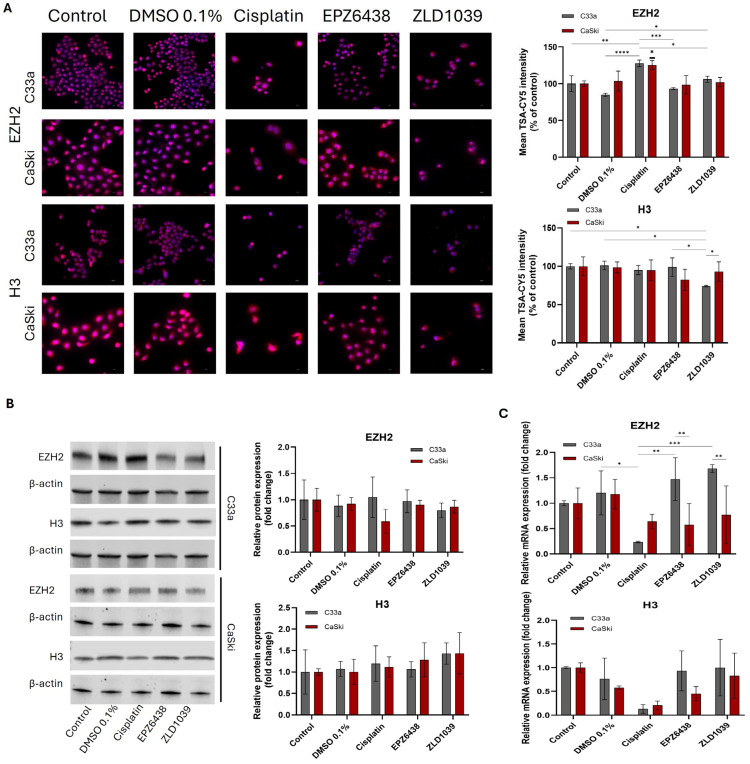
Effect of EZH2 inhibitors on expressions of protein and mRNA levels of EZH2 and H3 on cervical cells after 48 h treatment. (**A**) Immunofluorescence staining of EZH2 and H3 protein expression (pink) and nuclei (blue) under 400× microscope magnification. Scale = 50 µm. Bar charts on the right show mean fluorescence intensity of immunocytochemical staining results normalised to control cells (n = 3). (**B**) Representative Western blots’ results of EZH2 and H3 protein bands. Observed molecular weight is 93 kDa (EZH2), 17 kDa (H3), and 42 kDa (β-actin). Bar charts on the right showed Western blot quantification of EZH2 and H3 protein expression levels following treatment. Values are expressed as a fold change of control (n = 3). (**C**) RT-qPCR analysis of relative EZH2 and H3 mRNA expression levels following 48 h treatment. Values are expressed as fold change of control values. Data is presented as mean ± SD (n = 3). * *p* ≤ 0.05, ** *p* ≤ 0.01, *** *p* ≤ 0.001, **** *p* ≤ 0.0001. Bold asterisk shown above a single sample bar indicates the significant difference of that sample from the rest of the treatments.

**Figure 5 cimb-47-00990-f005:**
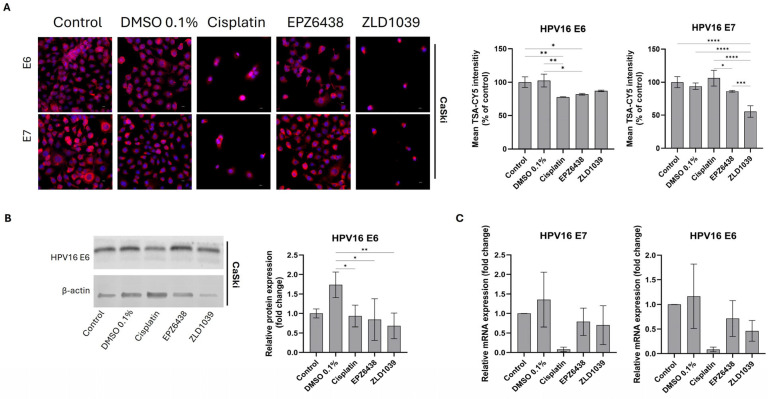
Effect of EZH2 inhibitors on protein and mRNA expression of HPV16 E6 and E7 on HPV+ cervical cells after 48 h treatment. (**A**) Immunofluorescence staining of HPV16 E6 and E7 protein expression (pink) and nuclei (blue) under 400× microscope magnification. Scale = 50 µm. Bar charts on the right show mean fluorescence intensity of immunocytochemical staining results normalised to control cells (n = 3). (**B**) Representative Western blots’ results of HPV16 E6 protein bands (**left**). Observed molecular weights are 17 kDa (E6) and 42 kDa (β-actin). Western blot quantification of HPV16 E6 protein expression levels following treatment (**right**). Values are expressed as a fold change of control (n = 3). (**C**) RT-qPCR analysis of relative HPV16 E6 and E7 mRNA expression levels. Values are expressed as fold change of control values. Data is presented as mean ±SD (n = 3). * *p* ≤ 0.05, ** *p* ≤ 0.01, *** *p* ≤ 0.001, **** *p* ≤ 0.0001.

**Figure 6 cimb-47-00990-f006:**
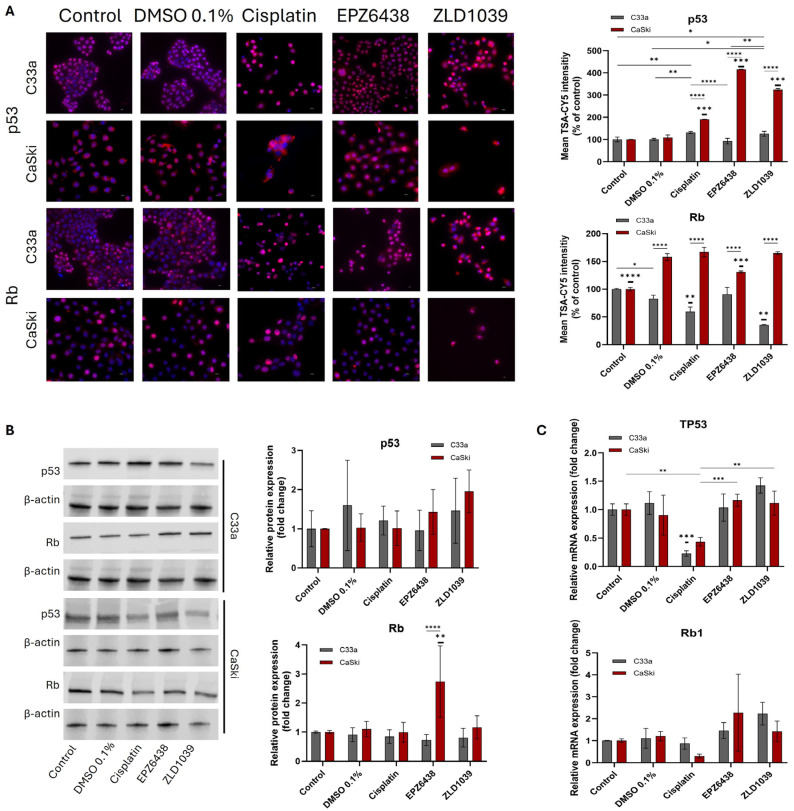
Effect of EZH2 inhibitors on protein and mRNA expressions of p53 and Rb on cervical cancer cells after 48 h treatment. (**A**) Fluorescence immunocytochemical staining of p53 and pRb protein expression (pink) and nuclei (blue) under 400× microscope magnification. Scale = 50 µm. Mean fluorescence intensity of immunocytochemistry results normalised to control cells is shown on the right (n = 3). (**B**) Representative Western blots’ results of p53 and pRb protein bands with observed molecular weights at 53 kDa (p53), 105 kDa (Rb), and 42 kDa (β-actin). Values of Western blot quantification are expressed as fold change of control (right) (n = 3). (**C**) RT-qPCR analysis of relative TP53 and Rb1 mRNA expression levels following 48 h treatment. Values are expressed as fold change of control values. Data is presented as mean ±SD (n = 3). * *p* ≤ 0.05, ** *p* ≤ 0.01, *** *p* ≤ 0.001, **** *p* ≤ 0.0001. Bold asterisk shown above a single sample bar indicates the significant difference of that sample from the rest of the treatments.

**Figure 7 cimb-47-00990-f007:**
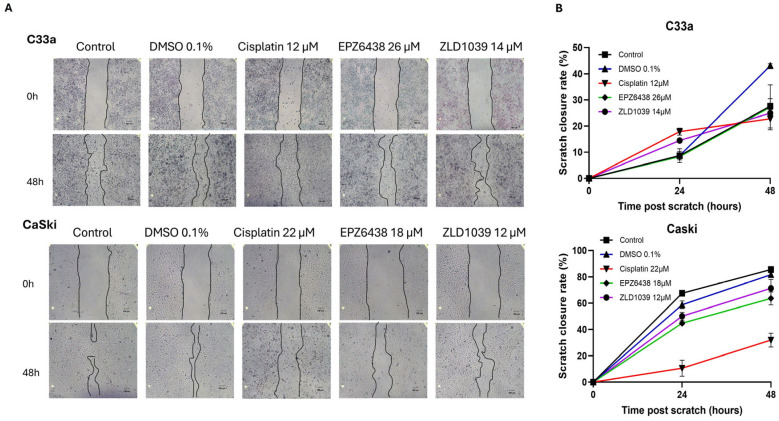
Migration rate assessment following the scratch wound–healing assay. (**A**) Representative images from the in vitro scratch wound–healing assays demonstrating cell migration into the cell-free region (outlined by black lines) following 48 h treatment (scale = 100 µm). (**B**) Summary plots showing the migration rates by C33a and CaSki cells after treatment (Mean ± SD, n = 3).

**Figure 8 cimb-47-00990-f008:**
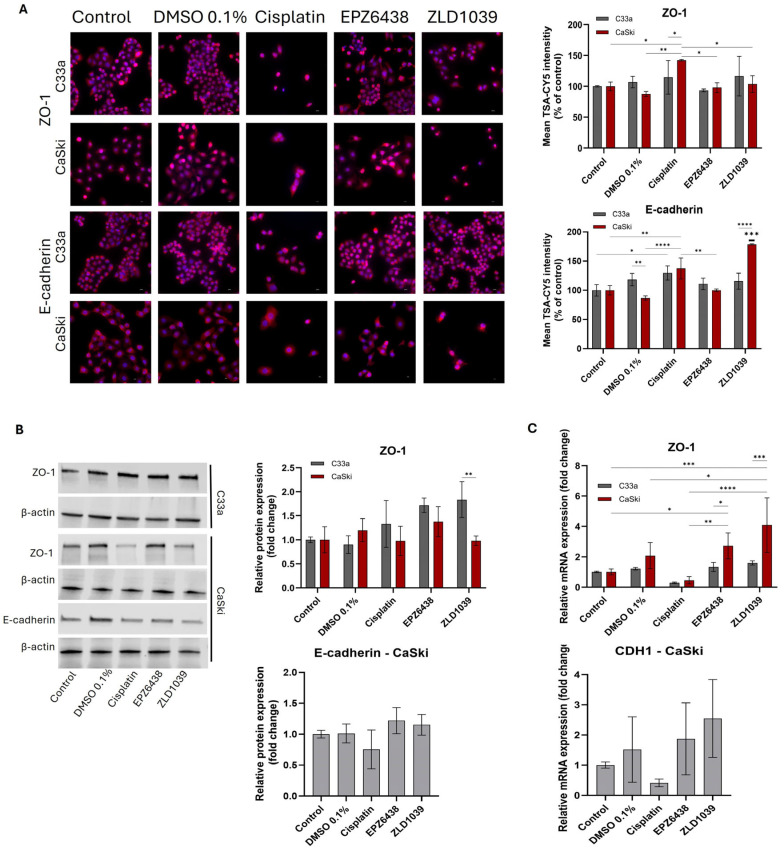
Effect of EZH2 inhibitors on protein and mRNA expression of ZO-1 and E-cadherin on cervical cancer cells after 48 h treatment. (**A**) Immunofluorescence staining of ZO-1 and E-cadherin protein expression (pink) and nuclei (blue) under 400× microscope magnification. Scale = 50 µm. Mean fluorescence intensity of immunocytochemistry results were normalised to control cells (right) (n = 3). (**B**) Representative Western blots’ results of protein bands ZO-1 (250 kDa) and β-actin (42 kDa) for C33a and CaSki and E-cadherin (110 kDa) for CaSki are shown on the left. Bar charts on the right show the Western blot quantification of these protein expression levels following treatment. Values are expressed as a fold change of control (n = 3). (**C**) RT-qPCR analysis of relative ZO-1 (C33a and CaSki) and E-cadherin (CaSki) mRNA expression levels following 48 h treatment. Values are expressed as fold change of control values. Data is presented as mean ± SD (n = 3). * *p* ≤ 0.05, ** *p* ≤ 0.01, *** *p* ≤ 0.001, **** *p* ≤ 0.0001. Bold asterisk above a single sample bar indicates the significant difference of that sample from the rest of the treatments.

**Figure 9 cimb-47-00990-f009:**
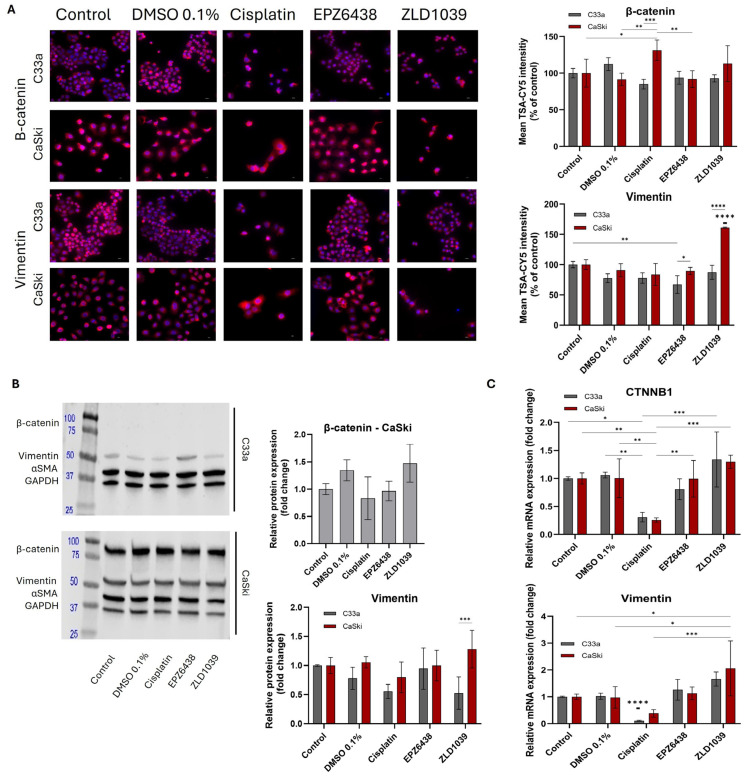
Effect of EZH2 inhibitors on protein and mRNA expression of β-catenin and Vimentin on cervical cancer cells after 48 h treatment. (**A**) Immunofluorescence staining of β-catenin and Vimentin protein expression (pink) and nuclei (blue) under 400× microscope magnification. Scale = 50 µm. Mean fluorescence intensity of immunocytochemistry results normalised to control cells (n = 3) is shown on the right. (**B**) Representative Western blots’ results of β-catenin and Vimentin protein bands (**left**). Observed molecular weights are 92 kDa (β-catenin), 54 kDa (Vimentin), 42 kDa (α smooth muscle actin), and 36 kDa (GAPDH). Western blot quantification of β-catenin and Vimentin protein expression levels following treatment (**right**). Values are expressed as fold change of control (n = 3). (**C**) RT-qPCR analysis of relative β-catenin and Vimentin mRNA expression levels following 48 h treatment. Values are expressed as fold change of control values. Data is presented as mean ± SD (n = 3). * *p* ≤ 0.05, ** *p* ≤ 0.01, *** *p* ≤ 0.001, **** *p* ≤ 0.0001. Bold asterisk above a single sample bar indicates the significant difference of that sample from the rest of the treatments.

**Figure 10 cimb-47-00990-f010:**
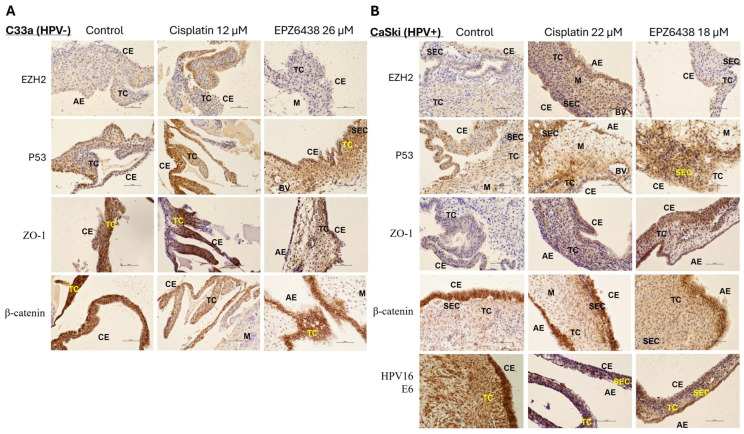
Immunohistochemical staining results of EZH2, p53, ZO-1, β-catenin, and HPV16 E6 from CAM assay tissue sections after 48 h treatment with control (DMSO 0.1%), cisplatin, and EPZ6438. Immunohistochemical staining of (**A**) C33a and (**B**) CaSki for expression of EZH2, p53, ZO-1, β-catenin, and HPV16 E6 from the CAM tissue model (n = 1). Scale = 50 µm. Chorionic (CE) and allantoic (AE) epithelial layers with sub-epithelial capillary network (SEC), tumour cells (TC), mesoderm (M), and blood vessels (BV) are displayed.

## Data Availability

The original contributions presented in this study are included in the article/Appendix A. Further inquiries can be directed to the corresponding author.

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
