# Peer review of "The Therapeutic Effect of EZH2 Inhibitors in Targeting Human Papillomavirus Associated Cervical Cancer"

_cimb, 2025, doi:10.3390/cimb47120990_

Round 1
Reviewer 1 Report
Comments and Suggestions for Authors
This manuscript presents a carefully designed and comprehensive study of the therapeutic potential of the EZH2 inhibitors EPZ6438 and ZLD1039 in HPV-associated cervical cancer. The authors successfully integrated various experimental approaches—cytotoxicity assays, flow cytometry, ICC, Western blotting, RT-PCR, and in vivo CAM validation—providing robust and compelling evidence. The finding that EPZ6438 demonstrates superior efficacy, particularly against HPV-positive cells, has scientific and clinical significance. Overall, the study is thorough, clearly structured, and makes a valuable contribution to cancer therapy. I recommend accepting it, however, I have two minor comments:
1) CAM assay results are based on n=1; replication or clarification of this limitation is recommended.
2) Line 358 - ZLD0139 instead of ZLD1039
Author Response
Many thanks for the reviewer's comments, here are the responses:
Comments 1: CAM assay results are based on n=1; replication or clarification of this limitation is recommended.
Response 1: Many thanks for the comments and we agree. We have revised this in discussion to emphasize this point. Specifically, we have now acknowledged this point in the discussion that the CAM assay results are preliminary based on a single experiment (n=1), which can be found at page 18, line 571-572 and 597. We also highlighted this as a limitation of the study, which can be found at page 20, lines 684-688. Due to time constraints and limited resources, we were unable to perform additional replicates at this stage, but we have clarified that repetition of this assay will be needed in future studies to verify these results.
Comments 2: Line 358 - ZLD0139 instead of ZLD1039
Response 2: Thank you for pointing this out. We agree with this comment. Therefore, we have corrected this spelling mistake at page 10, line 380.
Reviewer 2 Report
Comments and Suggestions for Authors
The study explores EZH2 inhibition (EPZ6438, ZLD1039) in HPV+ and HPV– cervical cancer models, combining in-vitro assays with a CAM in-vivo validation. The topic is timely and clinically relevant. Despite being an interesting manuscript, several points should be clarified:
Abstract:
- The Abstract overstates comparative efficacy versus cisplatin based on short-term viability and limited in-vivo data; temper the language to “show promise” and explicitly state the primary readouts (viability, apoptosis, cell-cycle, EMT marker expression).
- Add numerical effect sizes (e.g., % apoptosis increase, fold-changes in E6/E7, G0/G1 arrest, etc).
Introduction:
- The Introduction motivates EZH2 but the central mechanistic readout—H3K27me3—was not measured.
- Narrow the aim: specify hypotheses (e.g., “EPZ6438 reduces H3K27me3 and HPV E6/E7, restores p53/Rb, and shifts EMT markers toward epithelial state in HPV16+ cells”).
Materials and Methods:
- Cell lines and culture: Basic details are provided. Please add STR authentication dates and mycoplasma testing status. Clarify passage ranges used per experiment.
Cytotoxicity (MTT):
- Provide seeding justifications (50k cells/well in 96-well plates is high and can be confused with contact inhibition). Include calibration showing linearity of signal with cell number.
- IC50 was sometimes outside the tested range (EPZ6438 “could not be determined within 0–80 µM”); extend the range and/or fit robust models with constraints, and report confidence intervals.
- Treatments: Report final DMSO concentrations (vehicle) in all conditions; cap ≤0.1–0.2% and include vehicle controls at matched DMSO for every figure.
Apoptosis & cell cycle (flow):
- Detail gates/compensation, doublet discrimination, and the number of events analyzed (you note 10,000 in one place—confirm across experiments).
Scratch wound assay:
- Migration should be isolated from proliferation. Use mitomycin-C or serum deprivation and verify similar confluence at t=0; otherwise, slower closure may reflect reduced proliferation, not migration. If this process was performed in the experiment, cite it; if it was not performed and it is not possible to repeat the experiment, it should be included and discussed as a limitation of the study.
CAM assay:
- Include ethical oversight statement for avian embryos, or mention that a waiver from the ethics committee was obtained (if applicable).
Statistics:
- You mention two-factor ANOVA/Tukey or Kruskal-Wallis/Mann-Whitney, but with many endpoints the multiplicity burden is high; describe how familywise error is controlled (e.g., Holm-Bonferroni). Report exact p-values ​​and effect sizes (η², Cohen’s d).
Results:
- Viability: Dose–response trends are shown; however, drawing cross-drug efficacy comparisons (EPZ6438 vs cisplatin) needs matched pharmacodynamic metrics IC50 (e.g., Emax, AUC, Bliss independence in combinations). Also, explicitly provide IC50 ICs and replicate counts.
EZH2/H3 readouts:
- Key gap: no H3K27me3 measurement. Given EZH2’s mechanism, you should quantify global and/or locus-specific H3K27me3 by Western/ChIP-qPCR. The manuscript also notes inconsistency between H3 ICC and WB—discusses potential reasons (antibody specificity, epitope masking) and resolves with H3K27me3 (if possible).
HPV E6/E7 & p53/Rb:
- These are strong, relevant endpoints. Clarify whether C33A TP53 is mutant (it is widely reported), and interpret p53 accumulation accordingly. The Discussion mentions this discrepancy—moves some of that mechanistic clarification into Results with data if available.
EMT markers & migration:
- EMT panels are mixed (e.g., β-catenin undetected by WB in C33A; ZLD1039 increases some mesenchymal markers). Present full densitometry with statistics and discuss partial EMT states.
In-vivo CAM:
- Currently qualitative and underpowered (n=1). Provide quantitative tumor burden and IHC scoring, and replicate across embryos. Otherwise, frame as preliminary.
Discussion:
- Moderate claims about “superiority” to cisplatin; restrict to “under these in-vitro conditions” and avoid clinical extrapolation.
- Explicitly recognize the absence of H3K27me3 data and limited in-vivo replication as key limitations.
Conclusions:
- Reframe to “EPZ6438 shows promising preclinical activity and warrants further study,” avoiding efficacy superlatives or generalization to other HPV-associated malignancies without data.
Final Recommendation: Major Revisions.
Author Response
Many thanks to the constructive comments from the reviewer. Please find our responses below.
Comment 1: The Abstract overstates comparative efficacy versus cisplatin based on short-term viability and limited in-vivo data; temper the language to “show promise” and explicitly state the primary readouts (viability, apoptosis, cell-cycle, EMT marker expression).
Response 1: Thank you for pointing this out. We agree with this comment. Therefore, we have revised abstract accordingly at page 1, line 21-23.
Comment 2: Add numerical effect sizes (e.g., % apoptosis increase, fold-changes in E6/E7, G0/G1 arrest, etc).
Response 2: Thank you for pointing this out. We agree that including numerical effect sizes (e.g., percentage increases in apoptosis, fold-changes in E6/E7 expression, or the extent of G0/G1 arrest) would further strengthen the clarity of the findings. However, due to the strict 200-word limit for the Abstract, we were unable to incorporate these details without exceeding the allowed length.
Introduction:
Comment 3: The Introduction motivates EZH2 but the central mechanistic readout—H3K27me3—was not measured.
Response 3: Agree. We have, accordingly, revised this in discussion to emphasize this point. Specifically, we highlighted this as a limitation of the study, which can be found at page 20, lines 676-684.
Comment 4: Narrow the aim: specify hypotheses (e.g., “EPZ6438 reduces H3K27me3 and HPV E6/E7, restores p53/Rb, and shifts EMT markers toward epithelial state in HPV16+ cells”).
Response 4: Thank you for pointing thus out. We agree with this comment. Therefore, we have added the hypotheses of this study and amended relevant texts which can be found at page 3, line 133-139.
Methods:
Comment 5: Cell lines and culture: Basic details are provided. Please add STR authentication dates and mycoplasma testing status. Clarify passage ranges used per experiment.
Response 5: Thank you for pointing this out. We agree with this comment. We acknowledge the importance of STR authentication and mycoplasma testing for ensuring the validity and reproducibility of cell-line–based research. Unfortunately, STR authentication was not performed during the timeframe of our study. However, all cell lines were obtained directly from authenticated commercial sources and maintained under strict standard sterile culture conditions. In addition, we handled different cell lines separately when media were changed, and also cells were sub-cultured for each experiment we set up. We also generated a stock of vials containing the lowest passage number of cells to ensure that we use cells with the earliest passage number as possible and do not use those cells when they were passaged over 4 times. Mycoplasma testing was conducted regularly at every six months in our lab facility, and no positive results were found at the point of our work were carried out. Furthermore, to address concerns regarding potential HPV cross-contamination, HPV status for each cell line was confirmed by PCR, as presented in Supplementary Material S3. We have also added the passage number ranges used for each experiment in the revised Methods section (page 3, line 142; page 4, line 146; page 6, line 257).
Comment 6: Provide seeding justifications (50k cells/well in 96-well plates is high and can be confused with contact inhibition). Include calibration showing linearity of signal with cell number.
Response 6: Thank you for pointing this out. We agree with this comment. We acknowledge that the seeding density shown in the manuscript was incorrect, due to an editing oversight. The correct seeding density used in the experiments was 5,000 cells per well in 96-well plates (50,000 cells/mL with 100 µL added per well), and we have now corrected this in the revised manuscript (page 4, line 160). Regarding calibration, our assays were performed within the linear detection range (R2 value > 0.95). Therefore, a separate calibration curve was not included in the manuscript, however, all calibration data are available upon request. We agree that demonstrating linearity can improve clarity and have now added a statement to the Methods noting that MTT viability measurements were conducted within the assay’s linear range (page 4, line 160).
Comment 7: IC50 was sometimes outside the tested range (EPZ6438 “could not be determined within 0–80 µM”); extend the range and/or fit robust models with constraints, and report confidence intervals.
Response 7: Thank you for pointing this out. We agree with this comment. However, in our study, we did not further extend the concentration range for EPZ6438 for the non-cancerous cell line, as our primary objective was simply to confirm that the compound was not/ or less cytotoxic to normal cells within a biologically relevant range. For the cancer cell lines, the ICâ‚…â‚€ values were successfully determined and fell within the tested range: 26 µM for the HPV-negative cells and 18 µM for the HPV-positive cells. Given that our focus was on evaluating differential sensitivity rather than defining the full cytotoxic profile in non-cancerous cells, we did not proceed with additional testing beyond 80 µM. Nevertheless, we agree that extending the range or applying more robust modelling could provide a more precise estimate. Regarding confidence intervals, we have now reported this in results section at page 7, line 325-326.
Comment 8: Treatments: Report final DMSO concentrations (vehicle) in all conditions; cap ≤0.1–0.2% and include vehicle controls at matched DMSO for every figure.
Response 8: Agree. We have, accordingly, provided the final DMSO concentration as 0.1% to emphasize this point. This can be found in methods page 4, line 172. Furthermore, we have identified the typing errors from ICC image labelling for DMSO concentrations, and it has been all corrected now. Amendments can be seen at corrected Figures (3-6, 8-9), and page 17, line 548 and line 551(Figure 10).
Comment 9: Detail gates/compensation, doublet discrimination, and the number of events analyzed (you note 10,000 in one place—confirm across experiments).
Response 9: Agree. We have, accordingly, addressed these in the Methods section to emphasize this point. These details can now be found in Methods at page 4, lines 179-181 and 189-191, and in Figure 2 at page 9 line 359-360.
Comment 10: Migration should be isolated from proliferation. Use mitomycin-C or serum deprivation and verify similar confluence at t=0; otherwise, slower closure may reflect reduced proliferation, not migration. If this process was performed in the experiment, cite it; if it was not performed and it is not possible to repeat the experiment, it should be included and discussed as a limitation of the study.
Response 10: Thank you for pointing this out. We agree with this comment. While we did not apply mitomycin-C or serum deprivation, we monitored wound closure at an early 6-hour timepoint, during which proliferation is minimal. All cell lines demonstrated measurable wound displacement at 6 hours across all treatments, indicating true migratory activity. Because the 24- and 48-hour timepoints showed more pronounced differences, only these were included in the manuscript. We have now clarified this in the Methods at page 5, lines 198-202, and have added a note in the Discussion acknowledging that decreased cell proliferation could also play a role in slowing down wound closure, especially at later timepoints. This can be found at page 20, lines 689-693.
Comment 11: Include ethical oversight statement for avian embryos, or mention that a waiver from the ethics committee was obtained (if applicable).
Response 11: Thank you for pointing this out. We agree with this comment. The CAM model has been conducted using chick embryos within 14 days’ fertilization before hatching which no Home Office licence is required. However, this work was reviewed and approved by the local ethical committee of Natural Science department in Middlesex University and the laboratory risk assessment was completed prior to the commencement of the work. We have, accordingly, included this in manuscript at page 21, lines 753-754.
Comment 12: You mention two-factor ANOVA/Tukey or Kruskal-Wallis/Mann-Whitney, but with many endpoints the multiplicity burden is high; describe how familywise error is controlled (e.g., Holm-Bonferroni). Report exact p-values ​​and effect sizes (η², Cohen’s d).
Response 12: Thank you for pointing this out. We agree with this comment. Therefore, we have now substantially clarified our statistical methodology at page 7, lines 307-314. In addition to clarifying and expanding the statistical methodology in the Methods section, we decided not to embed extensive numerical statistical outputs (e.g., effect sizes for every comparison) directly within the Results in order to maintain clarity and narrative flow. Given the large number of endpoints, including all values in the main text would have disrupted readability. Therefore, full statistical outputs are available upon request. We believe that this would ensure a clear and smooth flow of the manuscript while all underlying quantitative data remain fully accessible at the same time.
Results:
Comment 13: Viability: Dose–response trends are shown; however, drawing cross-drug efficacy comparisons (EPZ6438 vs cisplatin) needs matched pharmacodynamic metrics IC50 (e.g., Emax, AUC, Bliss independence in combinations). Also, explicitly provide IC50 ICs and replicate counts.
Response 13: Agree. We have, accordingly, revised that in manuscript at page 7, lines 326-328 and page 8, line 342 to emphasize this point. We agree that cross-drug comparisons should be interpreted cautiously, therefore, texts were amended accordingly. The calculated IC50 values, maximal observed inhibition (Emax), and replicate numbers are now provided in Supplementary Table 4. Additional pharmacodynamic metrics (e.g., AUC or Bliss independence) were not measured.
Comment 14: Key gap: no H3K27me3 measurement. Given EZH2’s mechanism, you should quantify global and/or locus-specific H3K27me3 by Western/ChIP-qPCR. The manuscript also notes inconsistency between H3 ICC and WB—discusses potential reasons (antibody specificity, epitope masking) and resolves with H3K27me3 (if possible).
Response 14: Agree. We have, accordingly, revised this in discussion to emphasize this point. Specifically, we highlight this as a limitation of the study, which can be found at page 20, lines 676-684.
Comment 15: These are strong, relevant endpoints. Clarify whether C33A TP53 is mutant (it is widely reported), and interpret p53 accumulation accordingly. The Discussion mentions this discrepancy—moves some of that mechanistic clarification into Results with data if available.
Response 15: Thank you for pointing this out. We agree with this comment. Therefore, we confirm that the C33A cell line carries a TP53 mutation (as reported by ATCC), whereas CaSki cells have wild-type TP53. We did not directly measure p53 mutation status in our experiments and therefore do not have new data to report in the Results. However, we have addressed this in the Discussion (page 18 and 19, lines 616-631), noting the known mutation and suggesting that further assessment of p53 function would be valuable in future studies.
Comment 16: EMT panels are mixed (e.g., β-catenin undetected by WB in C33A; ZLD1039 increases some mesenchymal markers). Present full densitometry with statistics and discuss partial EMT states.
Response 16: Thank you for pointing this out. We agree with this comment. All Western blot quantification shown next to the representative bands reflects the averaged densitometry from three independent experiments, presented as fold-change relative to control, with error bars representing ±SD. Statistically significant differences (p < 0.05) are indicated with asterisks and original blots from all three independent experiments are provided in the Supplementary Materials. However, to clarify the partial EMT states further, we have added few sentences linking these quantitative observations to the partial EMT interpretation. These changes reinforce that the mixed marker patterns are biologically meaningful and consistent with previously reported hybrid EMT/MET states in cancer cells (page 19, line 657-668, and page 20 669-672).
Comment 17: Currently qualitative and underpowered (n=1). Provide quantitative tumor burden and IHC scoring, and replicate across embryos. Otherwise, frame as preliminary.
Response 17: Thank you for pointing this out. We agree with this comment. As these in-ovo measurements were performed once (n = 1), we presented them cautiously as preliminary observations (page 17, line 535; page 18, lines 571-572 and 597), and we highlighted in the Discussion that further replicate in-vivo studies are required (page 20, lines 684-688).
Discussion:
Comment 18: Moderate claims about “superiority” to cisplatin; restrict to “under these in-vitro conditions” and avoid clinical extrapolation.
Response 18: Agree. We have, accordingly, revised this in discussion to emphasize this point. Specific changes in moderation of wording can be found at page 20, lines 695 and 697, and page 21, lines 724-726.
Comment 19: Explicitly recognize the absence of H3K27me3 data and limited in-vivo replication as key limitations.
Response 19: Agree. We have, accordingly, revised this in discussion to emphasize this point. Specifically, we now acknowledge in the discussion that the CAM assay results are preliminary based on a single experiment (n=1), and we highlight this as well as absence of H3K27me3 data as a limitation of the study, which can be found at page 20 lines 676-688. Due to time constraints and limited resources, we were unable to perform additional replicates and H3K27me3 data, and we have clarified that repeating this in vivo assay and integration of H3K27me3 expression data will be needed in future studies to verify these results.
Conclusion:
Comment 20: Reframe to “EPZ6438 shows promising preclinical activity and warrants further study,” avoiding efficacy superlatives or generalization to other HPV-associated malignancies without data.
Response 20: Agree. We have, accordingly, revised this in conclusion to emphasize this point. Changes to moderation of this study findings can be found at page 21, lines 729-736.
Round 2
Reviewer 2 Report
Comments and Suggestions for Authors
The authors have thoroughly revised the manuscript and adequately clarified the key points raised by the reviewer. I recommend the manuscript for acceptance.